# Segregated Dynamical Networks for Biological Motion Perception in the Mu and Beta Range Underlie Social Deficits in Autism

**DOI:** 10.3390/diagnostics14040408

**Published:** 2024-02-13

**Authors:** Julia Siemann, Anne Kroeger, Stephan Bender, Muthuraman Muthuraman, Michael Siniatchkin

**Affiliations:** 1Department of Child and Adolescent Psychiatry and Psychotherapy Bethel, Evangelical Hospital Bielefeld, 33617 Bielefeld, Germany; msiniatchkin@ukaachen.de; 2Clinic of Child and Adolescent Psychiatry, Goethe-University of Frankfurt am Main, 60389 Frankfurt, Germanystephan.bender@uk-koeln.de (S.B.); 3Department for Child and Adolescent Psychiatry, Psychosomatics and Psychotherapy, Faculty of Medicine and University Hospital Cologne, University of Cologne, 50937 Cologne, Germany; 4Department of Neurology, Neural Engineering with Signal Analytics and Artificial Intelligence (NESA-AI), University Clinic Würzburg, 97080 Würzburg, Germany; muthuraman_m@ukw.de; 5University Clinic of Child and Adolescent Psychiatry, Psychosomatics and Psychotherapy, RWTH Aachen University, 52074 Aachen, Germany

**Keywords:** autism, biological motion perception, coherence, segregation, time-resolved Partial Directed Coherence

## Abstract

Objective: Biological motion perception (BMP) correlating with a mirror neuron system (MNS) is attenuated in underage individuals with autism spectrum disorder (ASD). While BMP in typically-developing controls (TDCs) encompasses interconnected MNS structures, ASD data hint at segregated form and motion processing. This coincides with less fewer long-range connections in ASD than TDC. Using BMP and electroencephalography (EEG) in ASD, we characterized directionality and coherence (mu and beta frequencies). Deficient BMP may stem from desynchronization thereof in MNS and may predict social-communicative deficits in ASD. Clinical considerations thus profit from brain–behavior associations. Methods: Point-like walkers elicited BMP using 15 white dots (walker vs. scramble in 21 ASD (mean: 11.3 ± 2.3 years) vs. 23 TDC (mean: 11.9 ± 2.5 years). Dynamic Imaging of Coherent Sources (DICS) characterized the underlying EEG time-frequency causality through time-resolved Partial Directed Coherence (tPDC). Support Vector Machine (SVM) classification validated the group effects (ASD vs. TDC). Results: TDC showed MNS sources and long-distance paths (both feedback and bidirectional); ASD demonstrated distinct from and motion sources, predominantly local feedforward connectivity, and weaker coherence. Brain–behavior correlations point towards dysfunctional networks. SVM successfully classified ASD regarding EEG and performance. Conclusion: ASD participants showed segregated local networks for BMP potentially underlying thwarted complex social interactions. Alternative explanations include selective attention and global–local processing deficits. Significance: This is the first study applying source-based connectivity to reveal segregated BMP networks in ASD regarding structure, cognition, frequencies, and temporal dynamics that may explain socio-communicative aberrancies.

## 1. Introduction

Autism spectrum disorder (ASD) is a psycho-neuro-developmental disorder characterized by deficient social interaction and communication as well as restricted, repetitive, and stereotyped behaviors and interests [1]. Disturbed imitation abilities are prevalent in individuals with ASD (for a review, see [2]) and correlate with language and communication abilities [3,4]. The underlying neural mechanism in ASD potentially rests upon impaired biological motion perception (BMP) [5] (for reviews, see [6,7,8]). BMP was first studied by Gunnar Johannsson in 1973 and describes the study of motion patterns in living beings using light points attached to their joints [9]. As neurotypical development indicates BMP to be innate, i.e., independent of early experience [10], a link to disrupted sociocognitive abilities in ASD is apparent [11]. At the neuronal level, individuals with autism (ASD) exhibit altered long-range connectivity patterns in regions normally associated with the mirror neuron system (MNS) [12]. However, there is conflicting evidence (for a review, see [13]) of connectivity between MNS and other networks. Abnormal BMP and inadequate MNS connectivity in autism may link basic processing deficits to ASD symptomatology [14].

### 1.1. BMP and MNS

Desynchronizations (suppression) of alpha (hereafter called mu (8–12 Hz)) and low-beta rhythms (13–20 Hz) observable in EEG over the sensorimotor cortex have been described as functional signatures of neurotypical BMP (e.g., [15,16]). Regarding mu, the literature points towards a clear definition of mu desynchronization overlapping with upper alpha rhythm in children and adolescents [6], especially during spatial attention [17]. Our main aim in this study was to look at distinguishing factors, namely the topographic specificity, spectral specificity, and functional dependence on behavior using source analyses and directed connectivity using time-resolved Partial Directed Coherence (tPDC).

Mu rhythm suppression covaries with BOLD signal changes in putative MNS [18], particularly with respect to connectivity patterns in response to biological kinematics [19]. MNS contributes to the conscious percept of humanness (anthropomorphism; see [20]) by converting abstract point-light stimuli into human-like figures [21]. Consequently, Mu suppression has previously been applied as a proxy for MNS activity (e.g., [22]) and is considered to index BMP processing. Furthermore, there is evidence for mirror neurons supporting social cognition, i.e., Theory of Mind (ToM) [23]. In line with this idea, participants with lower empathy were less accurate in identifying intentions in point-like walkers (PLWs) than in identifying the walker’s gender [24]. This provides a connection between BMP and ToM-like operations. That study found EEG generators in MNS towards BMP as well. We further assume that this suppressed EEG activity found in TDC also helps in identifying individuals with ASD.

### 1.2. Connectivity in Individuals with ASD

The structural integrity between frontal and subcortical structures deviates in various ways in individuals with ASD (e.g., [25,26], review by [27]). During brain maturation, there is a temporary short-range overconnectivity in early childhood and underconnectivity in later development [28,29] at lower frequencies [30].

Such deficient hardware settings also seem to affect functionality. Impaired structural connectivity involving MNS covaries with disorganized communication in imitation skills in ASD, affecting social and communicative symptoms [31,32]. Likewise, there is evidence for altered functional connectivity during BMP tasks in ASD [12]; while neurotypical connectivity patterns of BMP involve integrated form and motion networks [33], deficits of form processing were observed in individuals with ASD during BMP, including lower communication strength across the Superior Temporal Sulcus (STS) [34], lower connectivity within STS [35], and lack of integrated networks for form and motion located in the ventral (fusiform gyrus) and dorsal (parietal gyrus) stream, respectively. Low performance in response to BMP correlates with subclinical ASD symptom level [33] in healthy populations [36]. Furthermore, global form and motion processing is deficient in high-risk infants [37] despite preserved coherent motion perception per se in individuals with ASD [38]. This raises the possibility that inadequate BMP is specific to a lack of form and motion integration.

The present data regarding BMP processing in individuals with ASD are ambiguous. Although there is consensus regarding lower performance indices (see review by [39]), there is also evidence for neurotypical BMP in ASD (see, e.g., [40,41,42]). We identified several possible sources for such discrepancies. Some studies that reported results from small samples of young children that are more prone to individually different brain maturation phase ([42,43] compared PLW at different angles to assess task-difficulty effects and found neurotypical results for certain rotations in the ASD group. Meanwhile, the majority of results applies to adult samples [39], thus preventing a proper prediction of BMP to adolescent samples because of brain maturation processes. Underage ASD participants in [44,45] showed intact BMP regarding mu suppression, but slower [44] or less accurate [45] performance. Using a sophisticated study design, [46] reported comparable performance across samples in general, though collapsed over datasets, the ASD group performed below average. Unfortunately, no brain mechanisms were measured there.

Overall, there are methodological differences that may give rise to those contradictions [47]. There is no consistent definition of BMP tasks, e.g., meaningfulness of observed motion [48], self-executed vs. observed or imagined motion [49], human vs. non-human motion, or active vs. passive motion recognition [50]. Such task differences seem to affect BMP research on individuals with ASD, highlighting the need for simple BMP paradigms such as PLW [51]. Moreover, the underlying temporal dynamics are largely obscure. Identifying the direction of information flow both within MNS and towards associated regions (for form and motion) is crucial for the underlying functional connectivity [52] and could help in understanding ASD symptomatology. For this aim, we investigated BMP using EEG to identify the temporal interplay between core structures in this patient group.

### 1.3. Aim of the Present Study

We investigated neuronal networks involved in human motion perception in contrast to scramble motion in ASD vs. TDC. To operationalize abstract biological motion, participants passively watched proxies of human motion to obtain EEG measures and in a parallel version actively responded to the same percept for behavioral outcomes. BMP stimuli were generated using so-called point-light walkers, consisting of white dots tracking the movements of a real walking person via sensors at the joints of the limbs. Using EEG, we focused on coherent sources contributing to mu and low-beta rhythms, because PLW should desynchronize the mu and low-beta rhythms at central electrodes [24,53]. To characterize the dynamic interplay within this network, we applied dynamic imaging of coherent sources (DICS); this technique provides algorithms to calculate network correlations in source space for particular frequency oscillations [54,55,56,57]. Multivariate approaches to EEG-based directed connectivity are recommended to reveal causal relations between frequencies [58]. Following this, time-frequency causality estimation was realized via time-resolved PDC [57]. Finally, we looked at the potential role of mu and beta as well as the tPDC connectivity patterns as fingerprints for classifying ASD and TDC using SVM.

#### Hypotheses

(1)Behavior: In individuals with ASD, we predicted more difficulties in distinguishing human from random motion. We also explored brain–behavior correlations(2)EEG: Further, we postulated a weaker desynchronization in the mu and beta rhythms in response to BMP(3)Source Analysis: In addition, we anticipated altered network characteristics in individuals with ASD, possibly with distinct sub-networks for form and for motion(4)tPDC: We expected the associated coherence and directional connectivity values for mu and low-beta rhythms to be weaker and more local in individuals with ASD, with distinct directionalities between form, motion, and MNS(5)SVM: Finally, we anticipated that the behavioral and electrophysiological parameters can separate both groups (TDC, ASD) using classification algorithms.

## 2. Methods

### 2.1. Subjects

Children (*n* = 21) between 9 and 15 years old (mean age and standard deviation: 11.3 ± 2.3 years) with ASD and 23 age- and IQ-matched typically developing children (mean age 11.9 ± 2.5 years) were included (right-handed according to Edinburgh Handedness Inventory [59]; normal or corrected-to-normal vision). We measured male individuals with ASD only given the high prevalence in this gender group (see [17]) and chose the control group accordingly to consist of male children as well.

Patients with ASD were diagnosed according to ICD-10 [59] by experienced clinicians at the Department of Child and Adolescent Psychiatry of the Goethe University of Frankfurt (Germany), based on the German version of the Autism Diagnostic Observation Schedule (ADOS Modul 2-4, [60]) and the Autism Diagnostic Interview–Revised (ADI-R). The medication-free individuals with ASD sample included five children with childhood autism (F84.0), ten with Asperger’s syndrome (F84.5), and six with atypical autism (F84.1). TDC were recruited from local schools and screened with the German version of the Child Behaviour Checklist (CBCL, [61]) for any clinically relevant symptoms. Participants with T-scores > 60 on the second-order scales or >70 on any first-order subscales of the CBCL, respectively, were excluded.

Exclusion criteria for both samples were as follows: intellectual disability according to a standardized IQ assessment (percentile rank < 2 corresponding IQ < 70, respectively; for details see below), any neurological disorder (including epilepsy), preterm birth (<2000 g), any other comorbid psychiatric disorder, dyslexia, and psycho-pharmacotherapy. IQ was assessed using standard (above 11.5 years) or colored (below 11.5 years) progressive matrices [14]. The matrices test is a non-verbal, multiple-choice IQ assessment tool, which measures deductive reasoning.

All participants and their parents were informed about the study’s procedure and purpose, and written informed consent according to the Declaration of Helsinki was obtained from both the participants and their parents. All methods were carried out in accordance with the regulations of the Declaration of Helsinki and met the ethical guidelines for research including Good Clinical Practice in particular with minors. The study was approved by the Ethical Committee of the Medical Faculty of the Goethe-University Frankfurt

### 2.2. Experimental Design, Procedure, and Stimuli Presentation

The logical sequence of the applied methods is illustrated in Figure 1. A 30 × 37.5 cm flat-screen was placed 80 cm in front of participants’ faces. Application of a ‘chin-rest’ ensured a constant distance and minimized head movements. The room was dark during the entire experiment.

The human motion stimuli (‘walker’ condition) consisted of moving point-light displays without contours. Two male walkers were marked by 15 white dots at the joints against a black background, tracking movements at the joints of the limbs (conducted with Labview version 6; http//www.ni.com/labview). The walkers were shown in a frontal view walking with a speed of approximately two steps per second. Stimuli were based on motion capture data as previously described [62]. To extract the human part that contributes to BMP and isolate it from the basic perceptual portion, we introduced a baseline condition: The ‘scramble’ (i.e., baseline) condition was derived from the ‘walker’ condition as follows. After permuting the position of the 15 individual trajectories, the velocity profile along each trajectory was replaced with a constant velocity that was created by averaging the velocity over one cycle. The shapes of the single trajectories remained intact. This manipulation retains the overall frequency individually for each dot, but masks the acceleration profile indicative of biological motion.

The stimulus sequence was controlled by Presentation^®^ software (Neurobehavioral Systems, Inc., Berkeley, CA, USA, www.neurobs.com). The single ‘walker’ and ‘scramble’ versions appeared for 40 s twice in alteration and were distributed over two identical blocks, summing up to 80 s overall per condition (see Figure 2). Each individual stimulus was shown for 1 s in a randomized order once per block. Within a block, a white fixation-cross was presented for 2 s to separate the scramble and the walker condition. Between blocks, there was a potential break with durations varying between individuals. In order to disentangle biological motion recognition from motor responses, participants passively watched the motion stimuli and reported their impressions afterwards, which was not systematically noted and is not part of the analysis. To maintain participants’ concentration and supervise a constant fixation, one experimenter remained in the room. To obtain behavioral indices of BMP, the same procedure was repeated in the same experimental session (active task), where participants responded by assessing whether the observed motion was perceived as “walker” or “scramble”. The purpose of the experiment (that there was always either biological or scramble motion) was known to the participants in advance.

### 2.3. EEG Recordings

Continuous direct current EEG (DC-EEG) from 64 channels at a sampling rate of 500 Hz was recorded against a reference at FPz using BrainVision MR-Plus amplifiers and Brain Vision Recorder software 2 (Brain Products GmbH, Munich, Germany). An anti-aliasing low-pass filter with 250 Hz high cut-off was applied online. Sixty-four sintered Ag–AgCl electrodes (impedances <10 kΩ) were fixed on equidistant electrode caps (Easycap GmbH, Herrsching, Germany). Vertical and horizontal electro-oculograms (VEOG and HEOG, respectively) were recorded from electrodes placed 1 cm above and below the left eye and lateral to the outer canthi, respectively.

### 2.4. Signal Pre-Processing

EEG data were pre-processed in a first step using Brain Vision-Analyzer 2 (Brain Products, Munich, Germany). After offline average re-referencing, continuous recordings were segmented into epochs of 60 s using the following logic. As states of expectancy and awareness influence EEG oscillations in the mu range over the occipital cortex [63], the first and last 10 s of each 80 s block were eliminated. Thereby, possible attention transients associated with the initiation and termination of the stimulus [53,56] were minimized.

Subsequently, potential eye blinks and muscle artifacts were eliminated by a template matching two-step algorithm using CURRY (Neuroscan, Compumedics, Charlotte, NC, USA). To this end, a template containing the characteristics of a typical artifact is run over the whole data per channel to detect and remove analogous epochs matching the respective artifacts. This is done for each identified artifact. The EEG data set of each subject was parsed into ‘scramble’ and ‘walker’ conditions and exported to Matlab (The MatWorks Inc., Natick, MA, USA) for the subsequent analyses.

### 2.5. EEG Synchrony

In order to calculate the neural desynchronization index, the data were epoched from 1500 to 3500 ms around the start of each trial. A reference interval of −500 to 0 ms was used to calculate the percentage change between the active period (500–2500 ms) and this reference, using the classic method adapted from Pfurtscheller and Lopes da Silva [64]. Thus, desynchronization and synchronization are expressed as a percentage of activity relative to the reference interval. We used the first identified power source signal respectively for the mu and beta bands to estimate the neural desynchronization index.

### 2.6. Source Analysis

Our analyses concentrated on the frequency bands mu (8–12 Hz) and low beta (13–20 Hz). For source analysis thereof, the DICS source analysis from the fieldtrip toolbox [55,65] was applied to identify coherent brain sources contributing to the predefined frequency bands. DICS uses a spatial filter algorithm [66] and estimates tomographic power maps based on standard head models. The latter are derived from a more complex, five concentric spheres model [67,68] with a single sphere for each layer corresponding to the white matter, grey matter, cerebral spinal fluid, skull, and skin. Age-appropriate T1 templates in MNI space [69] served as basis for the volume conductor model. For the head model, the radius and the position of the sphere representing the standard electrode locations were applied. Finally, the lead field matrix containing the geometry and the conductivity information of the model was estimated from the described brain models.

To identify coherent sources, we first took the MNI co-ordinates of the identified clusters in each group separately. This allowed extracting the time series and estimating the corresponding source power (first source) separately for the two frequency bands (mu; beta). Next, we tested the source power with between-group *t*-tests (ASD; TDC). The source power was significantly (*p* < 0.01) different between the two groups for each source at both frequency bands. In addition, we computed the source coherence for each source separately and estimated significant differences between groups for each identified source irrespective of the groups (*p* < 0.01).

The resulting individual maps of the strongest cerebro–cerebral coherence were spatially normalized, averaged across subjects, and displayed on a standard MNI brain in SPM12 (Welcome Trust Centre for Neuroimaging). After identifying the coherent areas in the brain, the activity extracted from the surface EEG (sensor space) was projected into source space to obtain source activity. The original source signals for each source with several activated voxels were combined by estimating the second order spectra and employing a weighting scheme to form a pooled source signal for every identified source as previously described [70,71].

In all subjects, data from the ‘scramble’ condition formed the baseline for the statistical analysis of the condition of interest (‘walker’ condition). By comparing these difference-signals, effects of non-biological motion processing inherent in both conditions are expected to cancel out, isolating those brain systems specifically involved in the detection of biological motion [72]. Statistical analyses of non-directed coherence were based on group comparisons in source space. A grand average including all ASD and TDC subjects was computed for visualization purposes only.

### 2.7. Directionality Analysis

Using time-frequency causality, we can not only focus on a particular frequency itself, but can also analyze the time-dynamics of the causality at that frequency. Based on state–space modeling, the time–frequency causality estimation method of tPDC relies on dual-extended Kalman filtering (DEKF) [48,50]. This result in an estimate of the time-varying dependent autoregressive (AR) coefficients: One EKF estimates the states and feeds this information to the second EKF, which estimates the model parameters and back-propagates this information to the initial EKF. By concurrently using two Kalman filters working in parallel with one another, it is possible to estimate both states and model parameters of the system at each time instant. After estimating the time-varying multivariate (MVAR) coefficients, the next step is to use those coefficients for the calculation of causality between the time series. Since DEKF yields the time-varying MVAR coefficients at each time point, we can calculate tPDC at each time point as well. Afterwards, a time-frequency plot of all tPDCs concatenated produces a time-frequency plot. The precise distribution of the MVAR coefficients is not known; we used the surrogate method called bootstrapping [73] to check for the significance of the results. This method is based on randomly shuffling subjects’ time series and hence it is data-driven. In short, it divides the original time series into smaller non-overlapping windows and randomly shuffles the order of these windows to create a new time series. The MVAR model is fitted to this shuffled time series. This process is repeated 100 times and averaged. The resulting value forms the significance threshold value for all connections. This process is performed separately for each subject. Since the information flow between brain areas is difficult to estimate from EEG measurements due to the presence of noise and bias from volume conduction, any effective connectivity measure (here tPDC) has to be carefully tested for its reliability to detect the underlying neuronal interactions during any functional state of interest (here resting state). In this context, some authors use the imagery part of coherence [74] or time reversal technique (TRT) [75]. In a simulation study, Haufe and colleagues [75] demonstrated that the TRT is a suitable method to alleviate the influence of weak asymmetries (i.e., non-casual interactions caused by zero-lagged, instantaneous coherences (= volume conduction)) on the result of any causal measure, while maintaining or even amplifying the influence of strong asymmetries (i.e., time-lagged causal interactions not caused by volume conduction). Hence, TRT was applied as a second significance test on the connections already identified by tPDC using bootstrapping as a data-driven surrogate significance test. Accordingly, the tPDC asymmetries should be insensitive to contributions from volume conduction or other instantaneous interactions. In addition, our tPDC asymmetry calculation should completely revert by applying TRT, and therefore only be sensitive to strong causal interactions. We applied TRT on the tPDC values for each group (ASD; TDC) separately at both frequency bands and each single connection. Afterwards, the mean was estimated for each whole group separately (TDC and ASD) by taking all significant directed coherence values for further statistical analyses. The calculated means were compared with Kruskal–Wallis one-way analyses of variance tests.

### 2.8. Statistical Analysis

All statistical analyses were performed using SPSS version 14 (Chicago, IL, USA). Homogeneity between groups regarding age and IQ was statistically controlled for using z-tests. For the percentile ranks supplied for the cognitive level, a non-parametrical test for group comparison was conducted (Mann–Whitney U-test). Accuracy in the parallel active session was assessed using dprime (difference between standardized hits and false alarms). As normal distribution was violated in both conditions (dprime walker: Shapiro–Wilk = 0.837; *p* < 0.001; dprime scramble: Shapiro–Wilk = 0.87; *p* < 0.001), dprime was assessed using Mann–Whitney U-tests with group (TDC vs. ASD) as between-group factor. In both groups, brain–behavior correlations were further applied separately for hit rate and false alarm rate on the one hand and coherence and tPDC indices (separately for mu and beta) on the other hand using Pearson’s correlation coefficient (see Appendix A for a more detailed overview of the choice of significance thresholds). In both groups, there were several low-performing outliers with ≥1.5 standard deviations apart from the group mean dprime values (per condition). First, these were removed from the control group sample (*n* = 17) only, because low performance in the ASD group (*n* = 21) was predicted in Hypothesis (1) based on past findings showing a connection between ASD symptom severity and BMP performance [33,36]. In a second step, the same analysis was conducted on the ASD sample (*n* = 14) using the same cutoff (i.e., without the low performing outliers (≥1.5 standard deviations)) in order to validate the outcome (see Section 4.1 Behavioral Data for more details). The results of the tests were comparable.

For the electrophysiological data, age was included as covariate, and a within-subject surrogate analysis was used to test the significance of the identified sources. The basis for the surrogates was the Monte Carlo approach with random permutation (100 times shuffling) using one-second segments per subject. The *p*-value for each of these 100 random permutations was estimated to obtain the 99th percentile *p*-values as individual significance levels.

In order to identify group differences (ASD vs. TDC), the mean coherence values (i.e., total interaction strength between all sources) were estimated per subject. On these values, a Kruskal–Wallis one-way analysis of variance test was applied, separately for mean coherence and for mean directional coherence (tPDC). The significance level was set to *p* < 0.05.

### 2.9. SVM Analyses

SVM is a powerful tool [76] for non-linear classification between two data sets. In short, the algorithm looks for an optimally separating threshold between the two data sets by maximizing the margin between classes’ closest points. The points lying on the boundaries are called support vectors, and the middle of the margin is the optimal separating threshold. In most cases, the linear separator is not ideal, so a projection into a higher-dimensional space is performed where the data points effectively become linearly separable. Here, we used the third-degree polynomial function kernel for this projection due to its good performance and use the grid search (min = 1; max = 10) and examined 10 possible values and a heuristic value of gamma (0.25). A soft-margin classifier was used for every parameter, and misclassifications were weighted by a penalty constant C. In order to optimize classification accuracy, this was calculated for every classifier. The selection was checked by 10-fold cross-validation by taking 75% of the data for training and 25% for testing. The classification was conducted separately for each analyzed parameter of behavioral data, source power, coherence, and connectivity in both frequency bands. In order to have more robust testing phase, we repeated the analyses for each parameter for three different testing data samples for the SVM. In order to demonstrate that no overfitting is attested in our data for the SVM classification algorithm, we performed (and report only the results from) the SVM with 10-fold cross-validation. We have successfully applied SVM in previous studies for classification of two datasets [77,78].

## 3. Results

### 3.1. Sample Characteristics

There were no age differences between groups (t(42) = 0.86; *p* = 0.39). With regard to the cognitive level, TDC showed an average IQ percentile of 62.7% (±38.2), and children with ASD reached an average percentile of 76.2% (±27). There were no cognitive group differences regarding the percentile rank (z = 0.9; *p* = 0.36).

### 3.2. Behavioral Data

Regarding performance in the separately conducted active BMP task, there were several outliers (i.e., low hit rate or high false alarm rate) in the TDC group more than 1.5 SD from the group’s standardized mean. These were excluded from the main behavioral analysis. In the remaining data set (TDC = 17; ASD = 21), there was a significant difference between groups of accuracy (dprime) in both conditions (walker: *p* < 0.001; scramble: *p* < 0.001; (see Figure 3A).

To account for the fact that we also identified several outliers (≥1.5 SD) in the ASD group, we reanalyzed the data using *n* = 14, applying the same cutoff criterion (≥1.5 SD). The Mann-Whitney U-Tests remained significant (walker: *p* = 0.003; scramble: *p* = 0.015).

Due to the fact that we assessed not only performance but also neuronal aspects underlying BMP, and since we cannot narrow down the source(s) for the observed negative d-prime values, we included the whole sample for further neuronal analyses.

### 3.3. EEG Synchrony

In both examined frequency bands, the shift from baseline that reflects desynchronization was significantly stronger in the control group compared to the ASD group (mu: *p* = 0.0004; Cohen’s d = 0.91; beta: *p* = 0.003; Cohen’s d = 0.93) shown in Figure 4.

### 3.4. Coherent Sources

With respect to coherent sources in TDC, mu activity (Figure 5A) was represented by bilateral sources in the visual cortex (source 1), inferior temporal cortex (source 2), parietal cortex (source 3), inferior frontal gyrus (source 4), and putamen (source 5). Coherent sources generating mu activity in individuals with ASD (Figure 6A) were located in the bilateral visual cortex (source 1), posterior cingulate cortex (source 2), and thalamus (source 3).

Beta activity (Figure 5B) in the TDC group was associated with bilateral sources in the inferior parietal cortex (source 1), fusiform gyrus (source 2), insula (source 3), medial prefrontal cortex (source 4), and caudate nuclei (source 5). In individuals with ASD, bilateral sources in the inferior parietal cortex (source 1) and medial prefrontal cortex (source 2) as well as left post-central gyrus (source 3), and right hippocampus (source 4) elicited beta activity (Figure 6B).

All identified sources within subject surrogate analyses were statistically significant in the Monte Carlo random permutation test across all subjects per group. The sources identified in each subject survived the surrogate analyses with a significance threshold of (*p* = 0.002; Cohen’s d = 0.98). The mean coherence values (representing total interaction strength between sources) was stronger in TDC compared to ASD both for the mu (*p* = 0.0002; Cohen’s d = 0.94) and beta (*p* = 0.0003; Cohen’s d = 0.88) frequencies.

### 3.5. Directional Connectivity

The direction of information flow between sources for the frequencies mu and beta is shown in Figure 7 for sources identified in the TDC children and in Figure 8 for sources that were extracted in the ASD group. The histograms indicate strength of information flow between each of these sources. In general, the mean directional coherence was significantly stronger in TDC compared to ASD subjects for both mu (*p* = 0.023; Cohen’s d = 0.70) and beta (*p* = 0.0001; Cohen’s d = 0.99). In addition, fewer long-range connections were observable in individuals with ASD. This was also observable when estimating the Euclidean distance of the significant connections between the single sources (peak MNI) voxels) per group (see Appendix A).

In TDC, we found a dense connectivity between the sources in total (*n* = 10 in mu; *n* = 9 in beta band), whereas in the ASD group, the connectivity structure was less dense (*n* = 4 in mu; *n* = 5 in beta band). More anterior–posterior cortical connections were evident in TDC compared to ASD.

### 3.6. Brain–Behavior Correlation

Correlation analyses between behavioral indices and EEG data resulted in significant results for the baseline (scramble) condition only. In TDC, responses on scramble trials were associated with mu coherence in putamen (positive for hits, *p* < 0.05, R > 0.4; negative for false alarms due to walker response towards the scramble percept, *p* < 0.05, R < −0.4) and with beta coherence in medial PFC (negative for hits only, *p* < 0.05, R < −0.4). As well, mu directionality from inferior frontal gyrus (IFG) to visual cortex (tPDC) covaried positively when responses on scramble trials were correct (*p* < 0.05, R > 0.4) and negatively in the case of false alarms (*p* < 0.05, R < −0.4).

In individuals with ASD, hit rate on scramble percept trials was negatively associated with strength of beta directionality from left post-central gyrus to right hippocampus (*p* < 0.05, R < −0.4), while false alarm rate on those trials negatively correlated with that beta tPDC (*p* < 0.05, R > 0.5). See Table 1 and Appendix A for a correlation matrix explaining the chosen significant cases in more detail.

### 3.7. Classification Results

In the SVM analyses, source analysis parameters of the beta frequency showed higher values than those of the mu band in separating ASD from TDC; here we report the 10-fold cross validated accuracy for each tested parameter separately (see Figure 9).

First, the classification accuracy using mu source power as input was 83%, while SVM on the source coherence of mu led to 88% accuracy. In case of the beta band, source power was able to classify both groups with 89%, and for beta coherence it amounted to 92%.

Second, using the connectivity parameters as input, we found 87% SVM accuracy for the mu band and 93% for the beta band.

Finally, the best classification accuracy was achieved including all three parameters (power, coherence, and connectivity) in the respective frequencies, with an accuracy of 92% for mu and 97% for beta band.

We repeated the analyses using all three parameters (power, coherence, connectivity) for both mu and beta frequencies, which resulted in an overall accuracy of 95%. This means that the cumulative effect contributed by both frequency bands was somewhat lower than the effect of beta alone.

## 4. Discussion

The present study investigated ASD-related deficits in BMP and associated abnormal oscillations as well as the underlying connectivity strength and direction of the respective sources. To create BMP, we applied point-line human motion stimuli (walker condition) and randomly shuffled versions thereof (scramble condition). The scramble version was used as the baseline condition. EEG data from the TDC and ASD groups were recorded in a passive design, where participants observed each condition separated by blocks. Behavioral data were separately acquired using an active version without EEG. We focused on source analysis and the associated connectivity strength (coherence) and informed about the dynamics during motion processing using DICS. Essentially, our neurophysiological data point towards group differences at various levels including EEG synchrony, coherent source generator structure, and their communication paths. This applied to both investigated frequency bands (mu; beta) and correlated with performance indices of the control group in the baseline (scramble) condition.

### 4.1. Behavioral Data

The original sample (*n* = 44) consisted of several low performers that were identified by comparing each individual’s dprime value with the group’s mean performance level and defining a threshold of 1.5 standard deviations that we used as a cutoff. In a first analysis, we applied this criterion only to the TDC group, because we had expected low performance in the ASD sample. The adjusted statistical analysis yielded significant differences between ASD and TDC. The rationale for excluding TDC samples is derived from the assumption that healthy participants should in principle be able to differentiate between biological and scramble motion, whereas subclinical samples show low performance as well. To validate this outcome, we ran the same statistical test with the ASD sample outliers removed and found the group difference to be robust. This is in line with Hypothesis (1) derived from previous findings postulating performance differences between groups.

As we did not assess each individual’s developmental age level (as opposed to chronological age), it is possible that the outliers in both groups were at a developmental stage that exceeded their capability to detect biological motion. Although the experimental design was suitable to allow accurate observation, we cannot exclude the possibility that some participants responded only randomly because of low motivation. Supporting the observed behavioral differences, we found that BMP performance correlated with neuronal activity only in TDC, but with symptom severity in individuals with ASD. This suggests that TDC recruit more consistent, stable networks, whereas individual strategies could have been applied to facilitated BMP in ASD [33], which in turn shaped interindividually varying neuronal connections.

Regarding the neuronal underpinnings of these behavioral outcomes, we observed correlations only for motion perception per se, i.e., on scramble trials. In the neurotypical sample, performance covaried with mu coherence in the putamen (hits; false alarms) and medial PFC beta coherence (hits only), and directionality (i.e., tPDC) from IFG to visual generators correlated with the same performance indices in TDC. By contrast, tPDC from postcentral gyrus to hippocampus predicted ASD scramble trial responses in the beta range and walker hits for TDC in the mu range. This finding complements results from a recent meta-analysis of behavioral BMP studies pointing towards group differences that were not further related to any underlying neuronal mechanisms for lack of studies combining both methods [79]. Here, the brain–behavior implications can be further discussed below.

### 4.2. EEG

In line with research on BMP [15] and on MNS in particular [80], TDC showed desynchronization from baseline in the mu and beta ranges (Hypothesis 2). This observation was not present in the ASD group; one possibility is that they could not suppress internal self-directed reflections to engage in self–other reflections. The EEG sources support this argument.

### 4.3. Coherent Sources

Our study replicates the link between oscillations in the mu band with coordinated perceptual, sensory, and motor activation (e.g., [16]) hypothetically involved in MNS [81]. We found mu-related activity in a broad network comprising regions for basic perceptual, high-level sensory, and motor functions in TDC. We infer that we found signs of social cognition like ToM, which is crucial during BMP and enabled by mirror neurons [82]. Consequently, according to [83], MNS and ToM comprise coupled networks involved in communicative behaviors. By contrast, there was no sign of a higher-order network in the ASD group. Possible explanations might be group differences in basic perceptual abilities like global processing or more sophisticated cognitive operations of attention. Both were found to be aberrant in patients with global processing deficits who show traces of ASD already in the primary visual cortex [84], and ASD has been associated with attentional deviations mainly in frontal cortices [85]. Attention processes are part of BMP at various levels from identifying motion to recognizing intention [86]. However, the observed mu source in basic sensory (visual) regions contradicts visual processing abnormalities in the ASD group. Rather, the subcortical sources (thalamus; posterior cingulate cortex) seem to reveal internal-reflective processes [87]. These further overlap with the so-called Default Mode Network (DMN), which is active during reflective thinking [88], suggesting internal programs for self-generated reflection, possibly about the sensation of moving. This proposed internal mode of individuals with ASD stands in contrast to self–other differentiations during BMP [89], which predominantly activates inferior parietal structures [90]. In other words, individuals with ASD apparently neglected the point-like walker’s perspective and compared the self-reflected with the observed motion. Similarly, low mu frequency (10–12 Hz) was found to deviate in ASD with regard to topography and sources in response to imitated vs. imagined hand gestures [91]. Mu-related DMN sources together with the visual cortex are in line with segregated and parallel networks evoking mu oscillations during BMP in ASD.

MNS-associated structures were also observable in TDC at the beta band [53]. Higher-order integration of sensory and motor input by SPL has been reported previously [92] and may have enabled perspective-taking [93]. The present results further suggest an interplay between different specialized sensory networks, possibly via an integrating hub in midline structures (medial PFC; see [94]). As medial PFC oscillations reflect impulsive behaviors [95], more coherence possibly reflected a tendency to focus on the walker identification. Coherent MNS structures apparently assist in rejecting nonhuman (scramble) motion stimuli, which potentially interfered with the main task. Motion identification per se (as during scramble trials) is an intrinsic ability of the visual system early on [11], implying that the associated paths could be more consistent across subjects than higher-order skills such as biological motion (walker trials). Following this line, performance measures in scramble trials may likewise be more easily detectable and transferable to a motion detection network than finding a group-wise neural overlap for the more complex identification of biological motion.

In addition, the medial PFC is also involved in ToM [96], and joint activity of networks for ToM and MNS is necessary to mentally imitate observed motion. Such a dependency of ToM and MNS has been established previously, and MNS development in infants is a precursor for ToM maturation [97]. ToM abnormalities could explain deviating social development in individuals with ASD [98]. Unlike in TDC, beta oscillations in ASD originated in subcortical regions associated with somatosensation (postcentral gyrus) and memory retrieval (hippocampus). Deviations in BMP might be the result of a divergent development of already abnormal visual paths in ASD participants. In contrast to consciously perceiving humanness, individuals with ASD may have responded to BMP with automated motion simulation instead of thoughts about the abstract PLW. Like in TDC, there were beta oscillations generated in medial PFC. This group overlap in medial PFC additionally supports the idea of a supraordinate center receiving and distributing input in order to conduct complex social actions.

In contrast to these inferences, recent publications question the role of mu suppression in MNS altogether due to underpowered studies and methodological and basic conceptual issues [99]. Moreover, the brain regions giving rise to the mu band are widespread, including evidence for somatosensory instead of motor cortices [100]. Automatic imitation paradigms even seem to reveal intact MNS in ASD [101] independent of chronological age [40]. This implies that the MNS may not have been involved at all during our study task, which in turn questions the entity that gave rise to BMP differences in the present study. Alternatively, lower-level perceptual deficits may have impeded on a network of brain structures that ultimately led to performance differences between the groups.

Medial PFC and ToM suggest that BMP is linked with social communicative behaviors, possibly via the MNS. Yet, as there is no measure of social communication in this study, we cannot directly infer from the connectivity data on behavioral outcomes down to the individual level. We can only tentatively infer from group performance differences that BMP was aberrant in the ASD group. Finally, that the ASD profile yielded fewer significant sources than TDC leads us to suggest that they recruited a less consistent network (Hypothesis 3).

#### 4.3.1. Connectivity: Paths

Complementing EEG generators, directional analysis of biological motion networks in TDC and ASD confirm feedforward (posterior-to-anterior) and feedback (anterior-to-posterior) loops that differed between groups. Regarding tPDC, these paths were widespread in TDC but predominantly local in the ASD group (Figure 7 and Figure 8). The impact of the different numbers of sources between groups of course increased the likelihood of finding such a density difference of networks, which is critically reflected in the limitations section.

In TDC, we identified significant connectivity indices for mu visual via inferior temporal to parietal cortex (feedforward). This may mean that basic motion signals were forwarded to the visual cortex, which was behaviorally relevant in order to respond on scramble trials. In fact, there was also a correlation for hits (positive coefficient) and false alarms (negative coefficient) with connectivity indices from IFG to visual cortex. After that, feedback from putamen to parietal and visual cortex may represent mechanism by which the observed details were iteratively compared with the participants’ personal motion experience. This is especially likely regarding the simultaneous connectivity of putamen to the same occipital as well as parietal sources. Additionally, there were bidirectional connections (within frontoparietal cortices; between visuotemporal and frontoparietal cortices).

In the beta range, unidirectional pathways consisted of forward signals from precuneus to insula, possibly to enable a more elaborate comparison of the observed motion with a putative in sensomotion [102]. Feedback paths further connected the basal ganglia to visual sources, and bidirectional paths occurred between insula and medial regions as well as within visual areas. Bidirectional paths of MNS linking STS with sensory and higher-order brain regions are empirically validated [102]. STS has been associated with MNS particularly during imitation [103]. The results pattern in the TDC group also shows integrated activity of dorsal (visual–parietal) and ventral (visual–temporal) streams for form and motion and is in line with BMP literature. In sum, a dynamical circuitry of TDC in the mu range may have established coordinated exchange of information characteristic of an MNS. The complementary beta-related connectivity pattern reveals involvement of motor structures (basal ganglia) and suggests a switching role for medial brain regions. In particular, the insular source seemed to coordinate different motor-related signals simultaneously.

In individuals with ASD, connectivity in the mu range involved a circuit of visual via posterior cingulate cortex to thalamus and interplay between visual and thalamus regions. This network contrasts with TDC except for basic visual feedforward signals. The thalamus possibly coordinated activity between perception (visual) and DMN (post-cingulate). Moreover, the bidirectional exchange of signals between the thalamic and visual sources might reveal that the observed percept was translated into imitated motion illusion. This is in line with a suggested self-generated reflection triggered by BMP in ASD. Distinct from the TDC pattern, this imitation was probably not conscious and purposeful, but rather automatically triggered, as there was DMN activity in ASD but no task-positive network, for which there was evidence in TDC. Again, these results are coherent with the innateness of motion perception as previously discussed [11], i.e., ASD children primarily showed basic visual processing patterns. Accordingly, DMN activity in individuals with ASD seems to be accompanied by an inward-directed scaffold of interconnected brain regions that may be automatically triggered during passively viewing biological motion. Consequently, when confronted with stimuli that elicit activity in MNS in TDC, the participants with ASD seemed to remain at a DMN mode, focusing on internal motor operations without feedback loops that would provide an intersection between inner representations and external sensory input. Alternatively, individuals with ASD may struggle when shifting from an otherwise intact baseline mode that is inward-directed, i.e., DMN related activity, to an active mode containing the concept of others. Such a default mode interference has also been related to the comorbid clinical profile of ADHD in past studies (e.g., [104,105]). Thus, DMN incorporates structures not only relevant for social communicative actions such as ToM [106], perception of the self [88], or emotion recognition [107], which were all found this study in the ASD group, but it also overlaps with MNS [108].

Beta-related signals in ASD originated in inferior parietal cortex, spreading to postcentral gyrus. The latter apparently functioned as an intersection to exchange bidirectional medial and hippocampal input. Inferior parietal cortex is part of the biological motion network, in particular processing motion information [109]. Potentially, forward parietal signals observed here triggered procedural memory traces through coordinated signals between postcentral gyrus and hippocampus. The significant correlation between this unidirectional connection and scramble trial performance in individuals with ASD suggests that memorized movement was stressed at the cost of social information, in contrast to the sources located in MNS regions in TDC. In addition, the bidirectional connections between postcentral and cingulate structures could indicate inhibitory efforts to suppress motion execution, similar to those observed in TDC.

#### 4.3.2. Connectivity: Range

Regarding the overall range of connectivity, we observed a random structure in the ASD group with predominantly local networks [30]. Moreover, a lack of bidirectional tracts between frontal and parietal sources adds to past evidence of aberrant frontal-to-parietal connectivity in ASD [82]. Thus, abnormal brain development is characteristic of individuals with ASD, microscopically reflected in imbalanced inhibitory versus excitatory cell types, and ultimately favors local short-range over long-distance trajectories [110]. Evidence suggests that deviating maturation entails a random network scaffold in ASD, hampering a small-world architecture [111]. Such maldevelopment may favor detailed but segregated analyses, supposedly to achieve cost-efficiency of a network that is otherwise disadvantageous with respect to energy optimization [112]. Disintegrated short-distance avenues further precipitate in comprehensive but segregated perceptual evaluations found in individuals with ASD instead of local-to-global integration (Gestalt phenomenon, [113]), which possibly applies to BMP as well. The altered connectivity framework found in the present study therefore corroborates the idea of automatic self-monitoring in response to PLW stimuli, which elicited higher-order BMP only in TDC. By contrast, people with ASD may have taken the self as reference rather than extrapolating global information of BMP to a putative other. Deficient information integration from different sources is generally concomitant with ASD [82], and this superior role for details rather than Gestalt perception has been specifically linked with BMP [114]. Metacognitive operations pertaining to others’ motion observation represented by MNS structures and self-referencing has been associated with ToM activity previously, potentially enabling social-communicative actions [115]. Analogously, performance in biological motion tasks is impaired only in response to higher-order emotion recognition but not basic perception (e.g., [47,116]). For the present results, this implies that separate but intact networks for form and motion seem to be evident in ASD. Thus, there is also a lack of anthropomorphism phenomena in ASD [21], where humanness is attributed to form in dynamical motion; segregated processing avenues possibly impede on the temporal integration of form and motion information, which is essential for successful BMP [117].

#### 4.3.3. Connectivity: Strength

TPDC further informed about the strength of connections. Coherence values for mu and beta oscillations were significantly stronger in TDC than ASD. This finding corresponds to past reports of altered connectivity patterns in individuals with ASD that may cause a preference for details (e.g., ‘weak central coherence’ theory, [118]). This is consistent with the overall lower range in the ASD group noted above. It strengthens the assumption that BMP is aberrant in individuals with ASD, presumably due to a compromised integration of otherwise preserved low-level perceptual signals [12]. Moreover, this provides evidence for weaker representations of the PLW during BMP. Overall, weak connections in ASD between key structures for BMP add to past findings on ASD regarding inefficient small-world networks [119] and aberrancies of an MNS for BMP integration [5].

Putting all tPDC findings together, Hypothesis (4) applied in terms of local trajectories for individuals with ASD in both mu and beta frequencies that were further segregated without intersections. The outlined interpretations bridge a gap between physiological phenomena and symptomatology. We suggest that the participants with ASD were in an inward-directed mode during motion perception instead of inferring intention to the PLW, which is a key part of communicative behavior. Lack of extrapolation like that of Gestalt perception is therefore a likely cause that might explain why individuals with ASD fail to integrate gestures and mimic expressions into verbal context.

### 4.4. SVM Classifier

Classifying all factors using SVM, a putative ASD-specific pattern emerges that was predicted in Hypothesis (5) (Figure 9). Especially mu coherence and beta connectivity predicted ASD reliably. Mu oscillations may be particularly susceptible to deviations from stable states in individuals with ASD. By contrast, beta oscillations are apparently particularly vulnerable in terms of directionality in ASD. Because mu coherence represents a phasic baseline resting state [120] whereas beta oscillations are predominant in task-related states [121], the following explanations may be derived:

First, ASD is associated with a deviant baseline (DMN) activity or alternatively with problems when desynchronizing from this baseline. Second, individuals with ASD have difficulty responding to momentary demands when shifting from an internal baseline mode. Rigidity and reduced flexibility are neuropsychological fingerprints of individuals with ASD and essential components of the associated inherent stereotypic behaviors [122], to which a rigid or deficient (DMN) baseline might contribute. Regarding neuronal abnormalities, fewer long-range connections may cause individuals with ASD to persevere in an internal mode, in which external input automatically triggers self-directed programs instead of allowing active exchange of information. Lack of complex feedback loops as observed in TDC that involve long-range connections and more intersections further corroborate this.

Of note, when reanalyzing our data over both frequencies (see Section 3: Results), the effect was slightly lower. This circumstance reinforced our hypothesis according to which mu and beta contribute to separate networks.

## 5. Conclusions

As far as we know, this is the first study using source analysis and sophisticated coherence algorithms to study biological motion perception in individuals with ASD. Using this approach, it was possible to reveal distinct processing paths that may underlie potential social-communicative aberrancies in this clinical population. We were further able to successfully show different source activations for beta and mu with distinct interactions between the sources depending on the group factor. Based on this, we could characterize the topographic specificity and the spectral specificity in a spatial attention task in adolescents with or without autism.

In sum, we extracted generators for mu as well as beta that converged in overlapping networks in TDC. We speculate that integration in TDC is not restricted to sensory and motor input itself, but furthermore applies to joint activity of networks (MNS; ToM) within but also between different frequency bands (mu; beta). Therefore, our results in TDC suggest cohesive activity converging in complex feedback loops. BMP may hence be required to enable more sophisticated operations, ultimately realizing complex social interactions via orchestrated activity of overlapping networks at different frequencies.

By contrast, the observed results pattern in individuals with ASD hints at segregation at different scales from structure (coherent sources) to operations (motor; perception; cognition), time (frequency bands mu and beta), and dynamics (parallel feedforward streams). In addition, the results may point towards a lack of mu desynchronization (shift from DMN), which implies an internally directed state in the ASD group in comparison with the putative task-positive network in TDC.

Of note, by correlating BMP performance with neuronal activity, we could assess the implications of this abnormal brain activity and further show the extent to which this may affect overt behavior. Possibly, distinct paths between TDC and ASD develop at an early age such that internal states in ASD hinder the establishment of proper network of perceptual integration.

The results bridge a gap between findings of abnormalities in various cognitive domains reported separately for ASD so far. These include underrepresented long-range connectivity, poor higher-order recognition of PLW stimuli despite preserved simple perceptual capacities, deficient Gestalt processing, and lack of anthropomorphism. Our findings indicate that a central mechanism is an inefficient integration of different information streams.

Prospectively, assuming that MNS develops successively to enable ToM, PLW stimuli could serve as passive paradigms applicable at a young age to determine the risk for abnormal social skills. Future studies may directly correlate BMP, ASD symptoms, and neural traces to reconcile behavioral problems with neurodevelopmental changes. Clinicians may exploit this knowledge in the future both for diagnostics and therapeutic applications.

## 6. Limitations

This study applied an innovative and low-threshold, easily implemented task design to elicit BMP. Simultaneously, the baseline condition was matched in terms of low-level perceptual features including velocity and luminance. However, the passive nature of the task that was used for EEG precludes exact inferences between neurobiological processing and performance. Our deductions regarding compensatory efforts are thus speculative rather than data-driven. This applies to the finding of significant correlations of behavior and connectivity from postcentral gyrus to hippocampus in ASD, since that same tract itself was not significantly involved in the tPDC. In addition, our sample size was low (TDC: *n* = 23; AUT: *n* = 21) and further reduced due to several low-performers. This impedes on the generalizability of our conclusions, meaning also that the reported findings are not robust and require further studies to observe whether they are replicable. If so, the conclusions from the present EEG study may guide future studies focusing on the extent to which overrepresented local connections observable in individuals with ASD impede on conscious, overt processing of BMP. As our applied analysis methods are non-standard, we strongly recommend investigating the observed relation between BMP and connectivity in ASD further, potentially including brain imaging to contrast structural and functional connectivity as well.

Moreover, the fact that brain–behavior correlations were limited to the scramble condition makes it hard to clearly overlay the passive with the active experiment. In general, we had no strong hypothesis beforehand for the correlations that were explorative in nature, so our interpretations here must be handled cautiously. We tentatively suggest that during passive watching, the participants were more engaged in perception itself, whereas in the active experiment, participants were required to consciously reflect the PLW and translate this to an overt response. Possibly, correctness was more important for the TDC group than for the individuals with ASD. This in turn may have impeded on the single conditions to a different extent, as motion observation was probably relatively easy for both samples, but individuals with ASD may have needed more effort (and distinct brain pathways) than TDC to discern biological motion. Whether this was reflected in the passive condition as well cannot be determined. Additionally, it is possible that the explicit mentioning of biological motion in the instructions and of the passive condition elicited attention effects that are known to alter neural processing in individuals with ASD in particular [123]. In light of these findings, more complicated social messages (i.e., reporting the detected movement in the BMP condition) potentially affected the groups more (and with distinct consequences for TDC vs. ASD) than merely perceiving movement (which was the case in the scramble condition). Such differences between active and passive may therefore explain the observed correlation pattern. This alternative must be taken into account, as the observed mu and beta frequency bands overlap with those found for attentional processes as well [49].

Regarding methods, we applied EEG without concurrent structural imaging, but EEG has a limited spatial resolution regarding to-be-detected sources, particularly deeper generators, especially given the low number of sensors (64 ch). Essentially, this applies to our findings from caudate, putamen, and thalamus. However, newer studies could validate the results from the applied spatial filtering technique [124,125]. Therefore, we assume high accuracy of the applied DICS results; a recent study indeed confirmed comparability between scalp-based EEG source analysis and direct (invasive) electrocorticography. Sources based on subdural (ECoC) and scalp (EEG) measurements were both attenuated comparably due to the great impact of the SNR [126].

Another issue confounding the results is the fact that the number of derived sources differed between groups and thus may have led to a higher network density in the TDC group. This must be considered and weakens the argument of generally more dense networks in TDC than ASD. However, tPDC still hints at segregated and local communication in individuals with ASD, since the identified sources did not communicate via global long-range paths.

Concerning sample characteristics, our exclusion criteria preselected for moderate-to-high functioning volunteers with ASD (IQ ≥ 70), which may have created a ceiling effect for this group. If motion perception was only slightly affected in this group, then the PLW task may have been too easy to find more robust ASD differences in BMP. In fact, the samples were homogeneous regarding IQ as well as age (IQ *p* = 0.36; age *p* = 0.39). Additionally, the present groups were matched in terms of chronological rather than developmental age. This is advantageous for caregivers as well as public institutions such as schools, where individuals are also age-matched according to the traditional chronological standard. Imbalanced maturation of different cell types is inherent in individuals with ASD, and neural maturation may both shape and in turn be shaped by development. However, for research purposes, it would be interesting to assess BMP in ASD under developmental aspects rather than comparing them with age-matched controls, particularly since ASD is a neuro-developmental disorder.

Even though the study has many limitations, we think the electrophysiology-based analyses to look at specific dynamics in the topography and interaction between the activated sources can aid in devising future treatment strategies. Namely, for non-invasive stimulation treatments, a valuable option would be to select behavior-driven regions and time windows for stimulation to obtain optimal performance in minors with ASD. In addition, the applied task might also be used for training purposes and evaluate performance by electrophysiological measures such as directed connectivity between two regions.

## Figures and Tables

**Figure 1 diagnostics-14-00408-f001:**
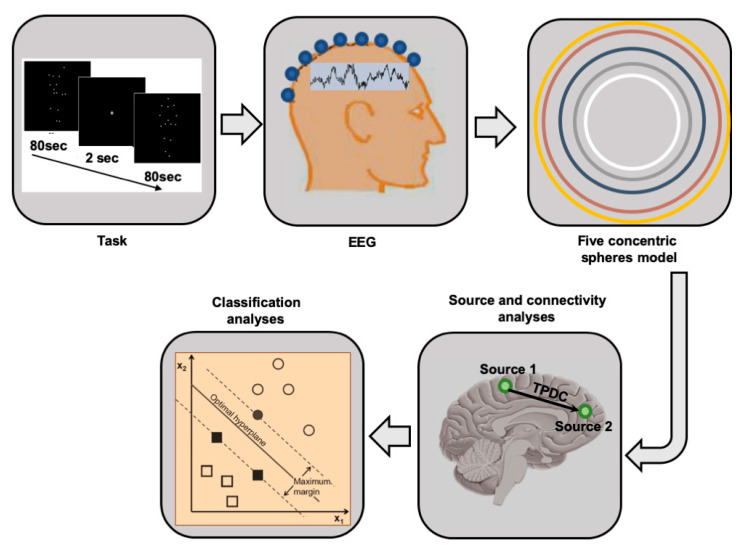
Overview of methods pipeline from data acquisition to source extraction and SVM.

**Figure 2 diagnostics-14-00408-f002:**
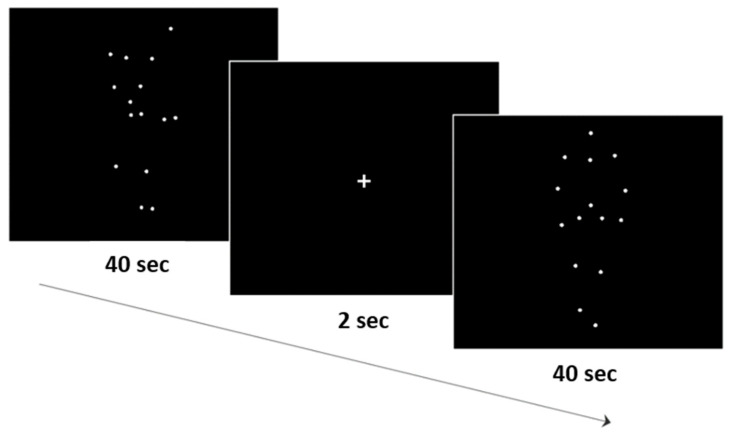
Display of one block. Stimuli were displayed as moving point-light displays without contours. The first ‘scramble’ condition appeared for 40 s, followed by a 2 s white fixation cross, and then the ‘walker’ appeared for 40 s. Order of presentation was randomized.

**Figure 3 diagnostics-14-00408-f003:**
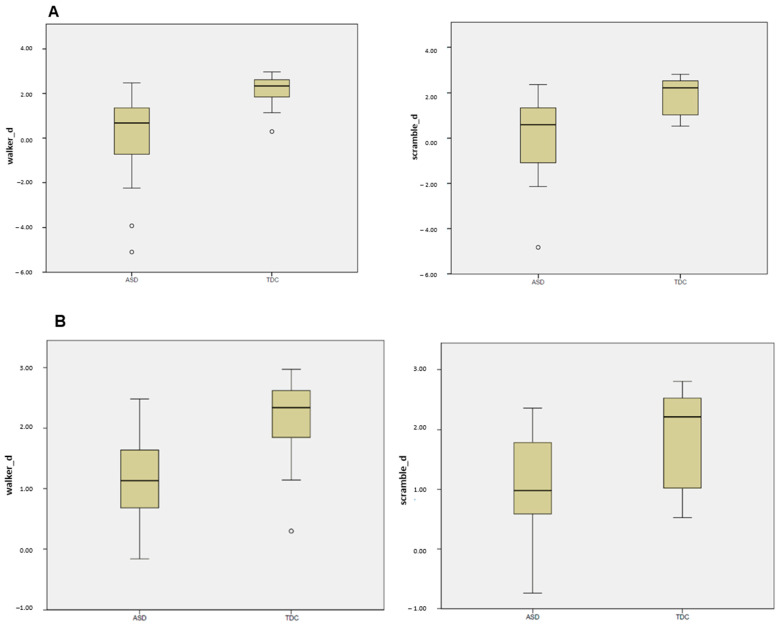
(**A**) Accuracy per PLW condition after removal of 1.5 SD outliers in the TDC group (left panel: walker = right panel = scramble) expressed as dprime (z-transformed hit rate vs. z-transformed false alarm rate), each for ASD (left columns) and TDC (right columns); outliers indicated by circles were less than 1.5 SD apart. (**B**) Accuracy per PLW condition after removal of 1.5 SD outliers in both groups (left panel: walker = right panel = scramble) expressed as dprime (z-transformed hit rate vs. z-transformed false alarm rate), each for ASD (left columns) and TDC (right columns); outliers indicated by circles were less than 1.5 SD apart.

**Figure 4 diagnostics-14-00408-f004:**
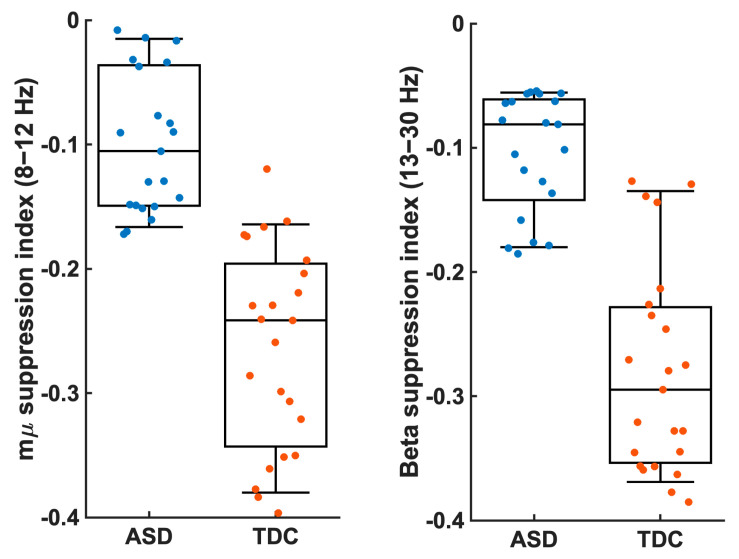
The mu suppression index and the beta suppression index were estimated for the first identified power source shown for the two groups separately (ASD (blue) and TDC (orange)). Suppression was significantly higher in TDC compared to ASD. (*p* < 0.01).

**Figure 5 diagnostics-14-00408-f005:**
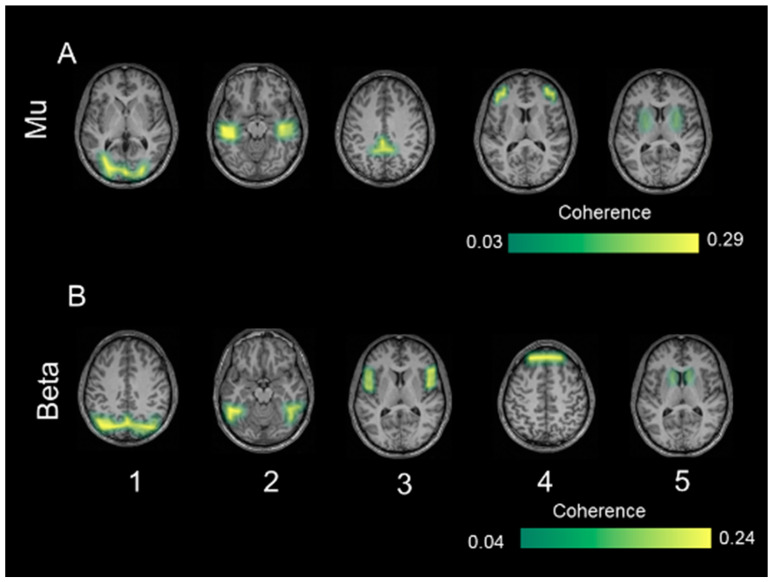
The identified coherent sources from typically developing controls separately for the mu frequency (8–12 Hz) in (**A**) and low-beta frequency (13–20 Hz) in (**B**). The first source indicates strongest power at each of the frequency bands respectively followed by the other identified significant sources. The color bar shows the significance threshold for identifying 0.05 for mu and 0.04 for low beta frequency.

**Figure 6 diagnostics-14-00408-f006:**
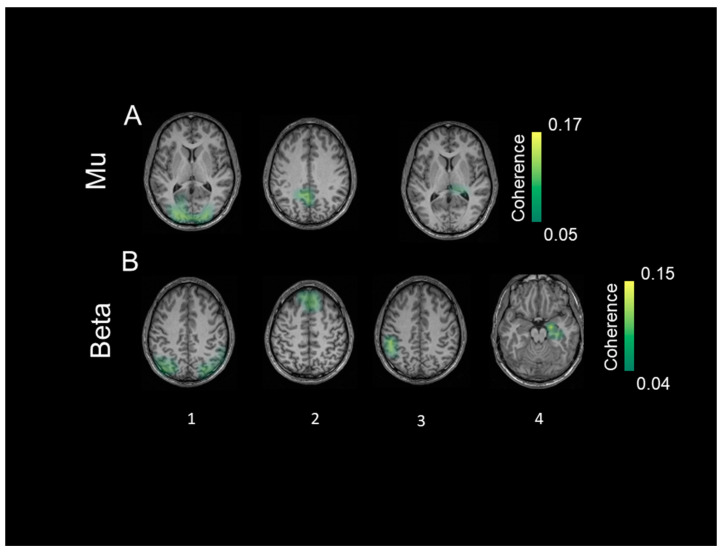
The identified coherent sources from the group with autism spectrum disorder separately for the mu frequency (8–12 Hz) in (**A**) and low beta frequency (13–20 Hz) in (**B**). The first source indicates strongest power at each of the frequency bands respectively followed by the other identified significant sources. The color bar shows the significance threshold for identifying 0.03 for mu and 0.04 for low beta frequency.

**Figure 7 diagnostics-14-00408-f007:**
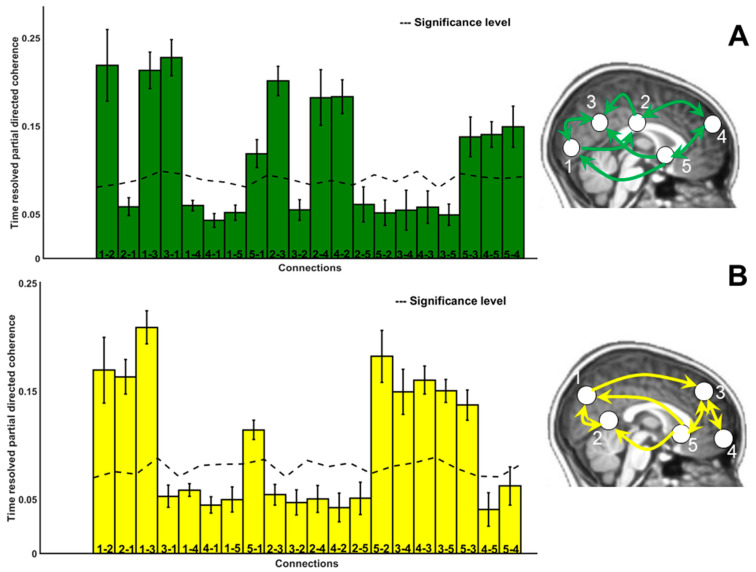
The direction of information flow separately for mu (8–12 Hz) in (**A**) and beta (13–20 Hz) (**B**) for TDC. Histograms indicate mean and standard deviation shown with error bars. The dashed line indicates the significance threshold, and significant connections are represented in the brain schematic with arrows.

**Figure 8 diagnostics-14-00408-f008:**
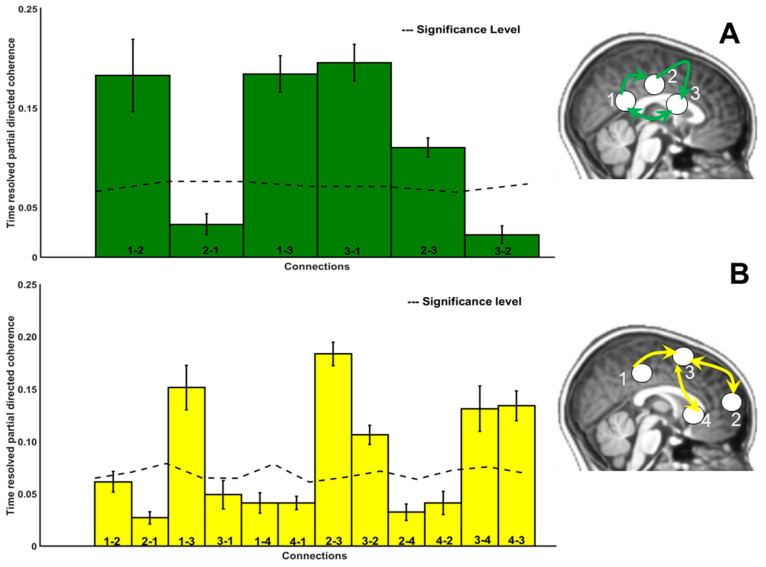
The direction of information flow separately for mu (8–12 Hz) in (**A**) and beta (13–20 Hz) (**B**) for ASD. Histograms indicate mean and standard deviation shown with error bars. The dashed line indicates the significance threshold, and significant connections are represented in the brain schematic with arrows.

**Figure 9 diagnostics-14-00408-f009:**
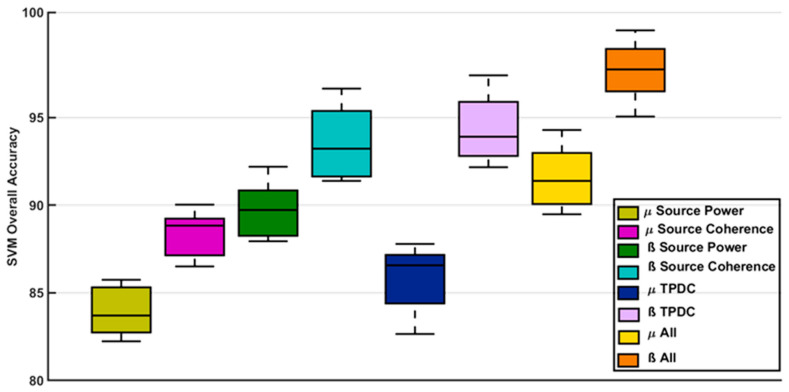
SVM overall accuracy (after 10-fold cross validation) separately for each of the following parameters: source power, mean source coherence (from all coherent sources), mean connectivity (TPDC), and all parameters combined (ALL). Box plots represent the variance in the 10-fold cross validation and separately for each frequency band, respectively.

**Table 1 diagnostics-14-00408-t001:** Significant correlations between indices (coherence; tPDC) of frequency bands (mu; beta) and behavioral outcomes (hit rates; false alarm rates) for both groups, respectively (TDC; ASD); only trials showing scramble motion were significant (HC = hippocampus; IFG = inferior frontal gyrus; mPFC = medial prefrontal cortex; n.s. = not significant; PCG = postcentral gyrus).

Scramble Percept			
**Group**	**Coherence**		
TDC		Mu (putamen)	
		hit rate	false alarm rate
	*p*	0.03	0.04
	R	0.46	−0.44
		Beta (mPFC)	
		hit rate	false alarm rate
	*p*	0.04	n.s.
	R	−0.43	0.37
	**tPDC**		
TDC		Mu (IFG to visual cortex)	
		hit rate	false alarm rate
	*p*	0.04	0.03
	R	0.43	−0.45
		Beta (PCG to HC)	
		hit rate	false alarm rate
	*p*	0.03	0.01
	R	−0.46	0.53

## Data Availability

Herewith we declare that all data can be made available upon reasonable request to the corresponding authors, who have moved to different locations working in separate labs. In particular, though data were acquired in Frankfurt, the more advanced analyses were conducted by other researchers not currently working there anymore. In case of interest, the algorithms can be explained and made available.

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
