# Peer review of "Segregated Dynamical Networks for Biological Motion Perception in the Mu and Beta Range Underlie Social Deficits in Autism"

_diagnostics, 2024, doi:10.3390/diagnostics14040408_

Round 1

Reviewer 1 Report

Comments and Suggestions for Authors

This study addresses biological motion perception deficits is children with autism spectrum disorder employing EEG methodology of source analysis and connectivity strength. The following comments and suggestions are offered with a hope to improve the quality of the manuscript.

1.     It would be helpful to have a summary at the outset of the Discussion including the experimental design, stimuli used, and data acquisition procedures along with the key EEG findings e.g., mu and beta oscillations, and coherent sources.

2.     It is unclear why outliers were excluded in the TDC group and how their exclusion affected the results.  Please provide more information on how low performers were identified.

3.     What may be broader implications/limitations of not assessing developmental age level?

4.     Please discuss the novelty/significance of correlations between BMP performance and neuronal activity.

5.     What is a rationale for the focus on scramble trials and how is it relevant to the study's goals?

6.      What are the implications of correlations found only for scramble trials in the neurotypical sample?

7.     Please expand on future research, addressing the present study's limitations and key findings.

Comments on the Quality of English Language

N/A

Author Response

Thank you for your kind review, we respondedto each point in the mauscript and added the answers in the reponse letter as well (see attachment):

Reviewer 2 Report

Comments and Suggestions for Authors

The results look very interesting for understanding the neuronal reasons of autism. The study was conducted at a high methodological level. Experimental designs, subject samples, signal processing, and analysis methods fully comply with all scientific research quality standards. The results are presented clearly; the conclusions are confirmed by experimental data. I have two minor remarks to improve the manuscript.

1. I recommend adding a brief definition of what is "BMP" at the beginning of the introduction (line 53). This definition will improve the understanding of the text.

2. In the manuscript, the terms "alpha-" and "mu-" rhythm are actually applied as the synonyms (line 54). In my point of view, this is not entirely correct. The mu rhythm coincides with the classical alpha rhythm in frequency, but differs in cortical topography and functional features. I recommend explaining more detail the difference between classical alpha and mu rhythms in the text of the manuscript.

Author Response

(The authors gave the same response as above.)
